

# Genomic signal processing for DNA sequence clustering

Gerardo Mendizabal-Ruiz, Israel Román-Godínez, Sulema Torres-Ramos, Ricardo A. Salido-Ruiz, Hugo Vélez-Pérez and J. Alejandro Morales

Departamento de Ciencias Computacionales, Universidad de Guadalajara, Guadalajara, Mexico

## ABSTRACT

Genomic signal processing (GSP) methods which convert DNA data to numerical values have recently been proposed, which would offer the opportunity of employing existing digital signal processing methods for genomic data. One of the most used methods for exploring data is cluster analysis which refers to the unsupervised classification of patterns in data. In this paper, we propose a novel approach for performing cluster analysis of DNA sequences that is based on the use of GSP methods and the K-means algorithm. We also propose a visualization method that facilitates the easy inspection and analysis of the results and possible hidden behaviors. Our results support the feasibility of employing the proposed method to find and easily visualize interesting features of sets of DNA data.

## INTRODUCTION

Cluster analysis is one of the most common and useful tools in pattern recognition, statistical data analysis, and exploratory data mining. It has many applications such as image segmentation, recognition of objects, document retrieval and others (*Jain, Murty & Flynn, 1999*). The main advantage of employing clustering techniques is the possibility of finding a hidden structure in the data without the requirement of prior information or knowledge about it. A clustering task consists of dividing a dataset into groups (i.e., clusters) that share common properties or that are related in some way, according to given criteria and similarity metrics (*Baikey, 1994*).

The most popular method used to perform cluster analysis is the K-means algorithm (*Jain, 2010*). K-Means clustering is an iterative partition technique which finds mutual exclusive spherical groups (*Joshi & Kaur, 2013*). The main advantage of the K-means algorithm is its ease of implementation and its linear time complexity (*Jain, Murty & Flynn, 1999*). However, the K-means algorithm rely on the frequent computation of similarity metrics between all of the elements to be clustered and the proposed centroids of each of the k-clusters. Therefore, its application in practice is limited to the type of data for which those similarity scores can be computed in a efficient way.

In bioinformatics, traditional methods for computing the similarity scores between sequences consist of applying DNA and amino acid sequence alignment methods, whose main objective is to identify portions of successive nucleotide or amino acids that are

Corresponding author
J. Alejandro Morales,
jalejandro.morales@academicos.udg.mx

common in two or more sequences. They are then rearranged to easily visualize those similar portions (*White et al., 2010*). The comparison of two sequences is known as pairwise sequence alignment (PSA). When more than two sequences are compared, the process is known as multiple sequence alignment (MSA) (*Sharma, 2008*).

One of the most popular applications of PSA is phylogenetic analysis. It consists of establishing an evolutionary relationship among nucleic acid or protein families sequences. It is generally depicted by the use of dichotomous trees, for which the branches represent organism separations. Branches that are close to each other, suggest a similar organism. By contrast, the farthest branches indicates large differences (*Mount, 2004*). Some of the most popular algorithms for MSA are ClustalW (*Thompson, Higgins & Gibson, 1994*), Muscle (*Edgar, 2004*), T-COFFEE (*Notredame, Higgins & Heringa, 2000*), MAFFT (*Katoh et al., 2005*), and K-Align (*Lassmann & Sonnhammer, 2005*).

However, since these methods require large computational times for determining similarity among sequences, the use of K-means is not feasible for this application. Therefore, other approaches for DNA clustering have been proposed based on the use of these similarity computation methods. Two of the most popular algorithms for clustering biological sequences are the CD-HIT (*Li & Godzik, 2006*) and the UCLUST (*Edgar, 2010*). Both algorithms use a greedy approach for identifying representative sequences that can be used as a "seed" to group all of the sequences that have a similarity score above a certain threshold. However, the computational resources necessary to perform the multiple sequence alignments remain the main challenge which limits the number of sequences that can be clustered.

More recently, an approach for the analysis of genomic data that has captured the attention of researchers in recent years, is the use of genomic signal processing (GSP) which is based on the use of digital signal processing (DSP) theory and algorithms to analyze DNA or protein sequences. GSP methods require the transformation or mapping of the biological sequences, usually represented as a string of characters (i.e., A, T, G and C) to a numeric representation (i.e., a signal) that can be processed using mathematical functions (*Kwan & Arniker, 2009*). Examples of the use of GSP methods include the identification of protein-coding regions in DNA sequences (*Das & Turkoglu, 2017*; *Mabrouk, 2017*; *Das & Turkoglu, 2015*; *Inbamalar & Sivakumar, 2012*; *Marhon & Kremer, 2011*; *Akhtar, Epps & Ambikairajah, 2008*; *Akhtar, Epps & Ambikairajah, 2007*; *Rushdi & Tuqan, 2006*; *Yin & Yau, 2005*; *Kotlar, 2003*; *Anastassiou, 2000*), finding for genomic repeats (*Sharma et al., 2004*), determining the structural, thermodynamic, and bending properties of DNA (*Gabrielian & Pongor, 1996*), biological sequence querying (*Ravichandran et al., 2010*), estimating of DNA sequence similarity (*Mendizabal-Ruiz et al., 2017*; *Hoang, Yin & Yau, 2016*; *Yin, Yin & Wang, 2014*; *Borrayo et al., 2014*; *Cheever et al., 1989*), and sequence alignment (*Skutkova et al., 2015*).

One of the main advantages of GSP methods is that the analysis of the genomic data can be performed very quickly because of the optimal coding of the algorithms and the processors that have been designed specifically for those tasks.

Cluster analysis of DNA signals through the use of GSP methods have been previously proposed by *Zhao, Duan & Yau (2011)* and *Hoang et al. (2015)*. However, these methods

are based on the computation of a number of features from the Fourier spectrum which may reduce the dimensionality of the data and perhaps its discriminative power as compared with the use of the whole raw spectrum. Moreover, those works employed a hierarchical clustering algorithm instead of the K-means. Comparatively, K-means properties allow us to generate plots that are different from the traditional dendrograms and that facilitate the exploration of the results.

In this paper, we propose an approach for performing cluster analysis of DNA sequences that is based on the use of GSP methods and the K-means algorithm. We also present a visualization method that allows us to easily inspect and analyze the results. Our results indicate the feasibility of employing the proposed method to find and easily visualize interesting features of sets of DNA data.

## MATERIALS AND METHODS

### DNA sequence to signal

In order to be able to employ the DSP methods in genomic data, it is necessary to first perform a transformation or mapping of the DNA sequences to be analyzed into numerical values representing the information contained by them. There are several proposed DNA numerical representations. However, one of the most popular of this DNA to signal mapping is the Voss representation, which employs four binary indicator vectors, each meant to denote the presence of a nucleotide of each type at a specific location within the DNA sequence (*Voss, 1992*).

Given a DNA sequence $\alpha$ (e.g., $\alpha = ATTCGCAT...$) we can employ the Voss representation to compute its corresponding fourth-dimensional DNA signal $\hat{X}^\alpha$ by applying Eq. (1)

$$
\begin{aligned}
\hat{\mathbf{X}}_1(i) &= \begin{cases} 1 & \text{if } X(i) = A \\ 0 & \text{otherwise} \end{cases} \\[1em]
\hat{\mathbf{X}}_2(i) &= \begin{cases} 1 & \text{if } X(i) = G \\ 0 & \text{otherwise} \end{cases} \\[1em]
\hat{\mathbf{X}}_3(i) &= \begin{cases} 1 & \text{if } X(i) = C \\ 0 & \text{otherwise} \end{cases} \\[1em]
\hat{\mathbf{X}}_4(i) &= \begin{cases} 1 & \text{if } X(i) = T \\ 0 & \text{otherwise} \end{cases}
\end{aligned}
\tag{1}
$$

By applying the Discrete Fourier transform to the DNA signal $\hat{X}^\alpha$, it is possible to compute the power spectral density (PSD) $\hat{S}^\alpha$ which describes how power of a signal or time series is distributed over frequency.[1] In our case, the PSD is a descriptor of the nucleotide patterns that may be present within the DNA sequence (*Borrayo et al., 2014*).

The relatedness or similarity score of any two given DNA sequences $\alpha$ and $\beta$, can then be estimated by comparing the components of their PSDs $d(\hat{S}^\alpha, \hat{S}^\beta)$ using a similarity metric (*Mendizabal-Ruiz et al., 2017*).

[1] For further details regarding the formal definition of a PSD refer to *Stoica & Moses (2005)*.

## DNA signal clustering

K-means is a two step algorithm which performs the partitioning of a given set of observations $\{O_1, O_2, \ldots, O_m\}$ represented as a $n$-dimensional vector, into $K \leq m$ clusters. Each cluster is represented by a centroid $C_j$ with $j \in [1, 2, \ldots, k]$, which is defined as a point in a $n$-dimensional space generated by computing the average of each element of the vectors of the observations that belong to that cluster. In the first step, an observation is assigned to the cluster $C_j$ that scores the highest similarity to the point represented by the observation's vector, according to a specific metric. In the second step, the centroids of the $k$ clusters are updated, according to the observations assigned to them in the previous step. The best groups and their centroids are obtained by the minimization of the total sum of the distances between the observations and their corresponding centroids.

Consider a set of PSD $\Omega = [\omega_1, \omega_2, \ldots, \omega_m]$ corresponding to a number $m$ of different DNA sequences. The K-means algorithm is applied to the data in $\Omega$ by considering the power spectra as the vector that describes the DNA sequence in a $n$-dimensional space. In this work, we chose the Euclidean distance between these vectors as the similarity metric to be employed by the K-means algorithm. Since the K-means results depends on the initial labels assigned to each entry, which are assigned randomly, we repeat the computation 50 times and keep the best convergence score. As a result, we obtain a label for each element of $\Omega$ which defines the assigned cluster.

## DNA clusters visualization

The raw results of the clustering procedure may be difficult to analyze and interpret. Therefore, we propose to produce graphical representations of the results that can easily provide an insight into the DNA sequence clustering results. The generation of the proposed graphical representation (Fig. 1) from the K-means clustering result, consists of the following steps:

1. Compute a main centroid point $M$ in the $n$- dimensional space corresponding to the geometrical center of the $K$ centroids location computed as:

$$M[i] = \frac{1}{k} \sum_{j=1}^{k} C_j[i] \qquad (2)$$

   where $i \in [1, 2, \ldots, n]$.

2. For each cluster $j$, compute the Euclidean distance $d_j$ of its centroid $C_j$ with respect to the main centroid $M$:

$$d_j = \sqrt{\sum_{i=1}^{n} (C_j[i] - M[i])^2}. \qquad (3)$$

3. Each centroid of the $k$ clusters is sorted according to its distance to the main centroid and an angle is assigned to them, according to its index $\iota \in [0, 1, \ldots, k]$ in the sorted array:

$$\theta_\iota = \iota \frac{2\pi}{k}. \qquad (4)$$

4. The main centroid $M$ and the clusters centroids $C_\iota$ are mapped into a two dimensional space $\phi$, where the main centroid corresponds to the origin (i.e., the point with coordinates $(x = 0, y = 0)$).

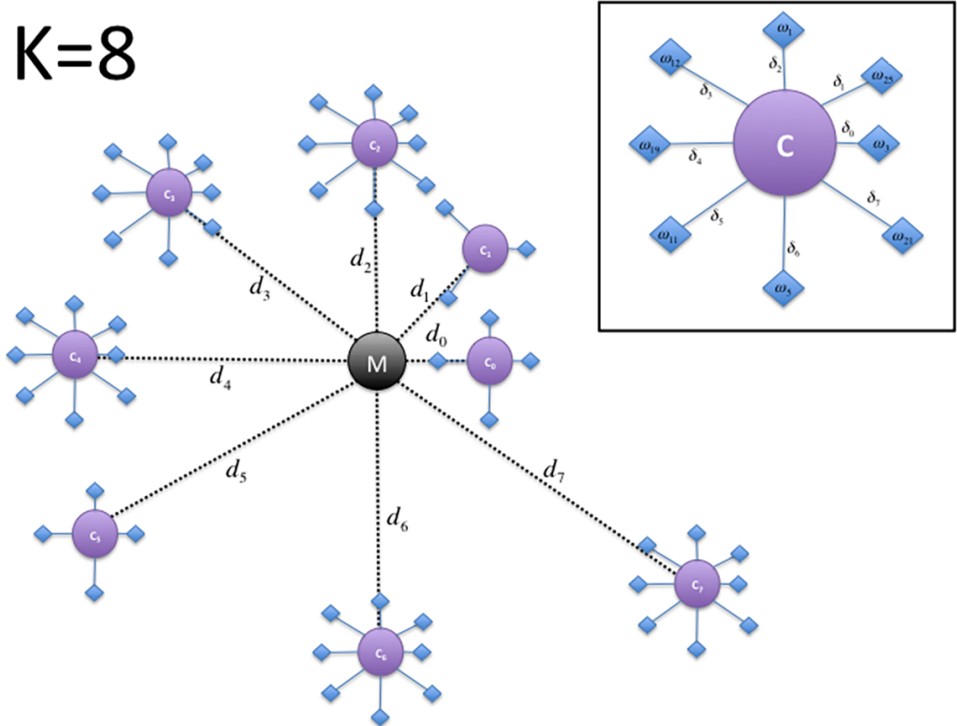

**Figure 1** Depiction of the DNA cluster visualization results structure proposed plot for a value of $k = 8$.

5. Each centroid $C_\iota$ is plotted as a point around the main centroid $M$ point, according to its distance and its angle as computed by:

$$x_\iota = d_\iota \cos(\theta_\iota) \qquad (5)$$

$$y_\iota = d_\iota \sin(\theta_\iota). \qquad (6)$$

6. We sort each set of DNA sequences in $\Omega$ assigned to a specific centroid $C_\iota$, according to the distance $\delta_z$ of each sequence $z$, with respect to its assigned centroid. The angle $\theta_z$ is also computed similarly to step 3.

7. Finally, each sequence $z$ is then plotted into $\phi$ by computing their correspondent coordinates as:

$$x_z = \delta_z \cos(\theta_z) + x_\iota \qquad (7)$$

$$y_z = \delta_z \sin(\theta_z) + y_\iota. \qquad (8)$$

## Experimental data

To assess our DNA signal clustering method and the proposed visualization technique, we employed a set of 141 DNA sequences corresponding to the Cytochrome c oxidase I gene (COXI) marker belonging to 131 different species obtained from the Kyoto Encyclopedia of Genes and Genomes (KEGG) K02256 (*Kanehisa et al., 2017*; *Kanehisa et al., 2016*; *Kanehisa & Goto, 2000*). We selected the COXI marker because it performs a fundamental role in the terminal oxidative step for energy metabolism (*Adkins & Honeycutt, 1994*) and is a very well known marker commonly used for the identification

of species (*Patwardhan, Ray & Roy, 2014*). In the selected set, a total of 112 organisms have only one copy. However, there are some species represented by more than one sequence. This is the case of the Yak, *Bos grunniens* (bom:102267288, bom:102278784, bom:22161768), a Bat, *Myotis davidii* (myd:102771221, myd:22203924), the Spotted green pufferfish, *Tetraodon nigroviridis* (tng:BAE79219, tng:GSTEN00036010G001), the Pacific giant oyster, *Crassostrea gigas* (crg:109618508, crg:109618509, crg:808829), *Yarrowia lipolytica* (yli:YalifMp03, yli:YalifMp05, yli:YalifMp06), *Loa loa*, the parasite responsible for filariasis disease (loa:COX1, loa:LOAG_19059), the Castor oil tree, *Ricinus communis* (rcu:10221395, rcu:8272741), and the Picoplanktons, *Ostreococus tauri* (ota:OstapMp24, ota:OstapMp40), *Bathycoccus prasinos* (bpg:BathyMg00110, bpg:BathyMg00240), and *Micromonas commoda* (mis:MicpuN_mit45, mis:MicpuN_mit7). It is important to note that all gene copies were considered during the experiments and that the selected organisms belong to the total spectrum of the Eukaryote domain.

The selected organisms were manually organized according to their respective taxon, based on the Catalogue of Life (*Roskov et al., 2017*), and were divided into seven kingdoms, 17 phyla, and 35 classes. To easily identify the different categories, we employed different colors and symbols as described in Fig. 2

## RESULTS

We employed the proposed method to evaluate how the experimental dataset is clustered, when using different values of $k$. While there may be many different criteria to select the number of clusters to be employed, in this work, we examine the results that are obtained employing three values that we consider interesting: (i) $k = 6$ which correspond to the number of different kingdoms in the selected dataset, (ii) $k = 17$ which corresponds to the number of different phyla in the dataset, and (iii) $k = 35$ which corresponds to the total number of classes in the selected dataset. The length of the PSD of each sequence to be compared was 4,100.

Figure 3 depicts the results obtained when the dataset was grouped into six clusters.

It can be noted that the majority of the Chordates (blue-edge squares) are grouped together in C-2, a small proportion of them (the Bat *Myotis davidii*, two copies of the Yak *Bos grunniens* and the second copy of the Spotted Green Pufferfish *Tetraodon nigroviridis*, are grouped in C-4 along with a plant (the second copy of the Castor oil tree *Ricinus communis*), and the rest are scattered in C-3 and C-6. It is remarkable that all of the Tracheophytan plants (green-edge squares) with the exception of Castor Oil tree *R. communis* are grouped in C-1 while the Chlorophyta plants (mostly Picoplanktons defined by green-edge circles) are grouped in C-6 along with other organisms. The formation of two separated groups for the plants may be explained by the fact that despite belonging to the same kingdom, these phyla share very little morphology (*Simpson, 2006*). Note that the all of the Ascomycota fungi (yellow-edge squares) are grouped together in cluster C-3, while the single organism of the Basidiomycota phylum (yellow-edge circle) is clustered in a separate group.

All Arthropods (blue-edge right oriented triangles) are contained in the cluster C-5. Note that the length of all of the organisms with respect to the centroid of the cluster
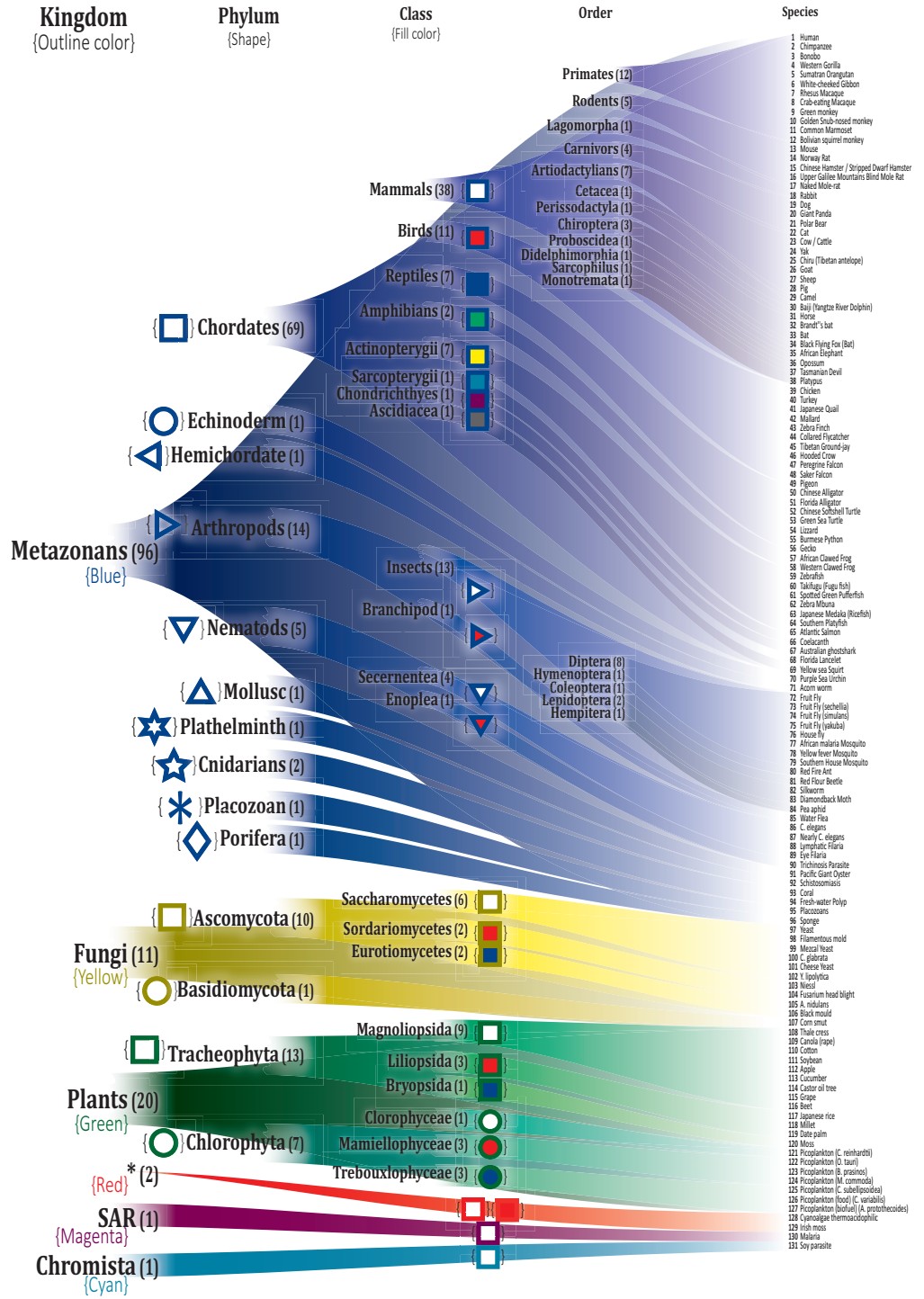

**Figure 2 Depiction of the selected organisms and their correspondence in the Tree of Life.** The respective hierarchic markings for each class is shown next to them. A detailed list of names and their KEGG entries is in Table S1. *These two organisms *Galdieria sulphuraria* (gsl:JL72_p19) and *Chondrus crispus* (ccp:ChcroMp03), a Cyanoalgae thermoacidophilic and Irish moss, respectively, do not have a reported Kingdom in the Tree of Life and were reported with the same Kingdom label 'Unknown'.

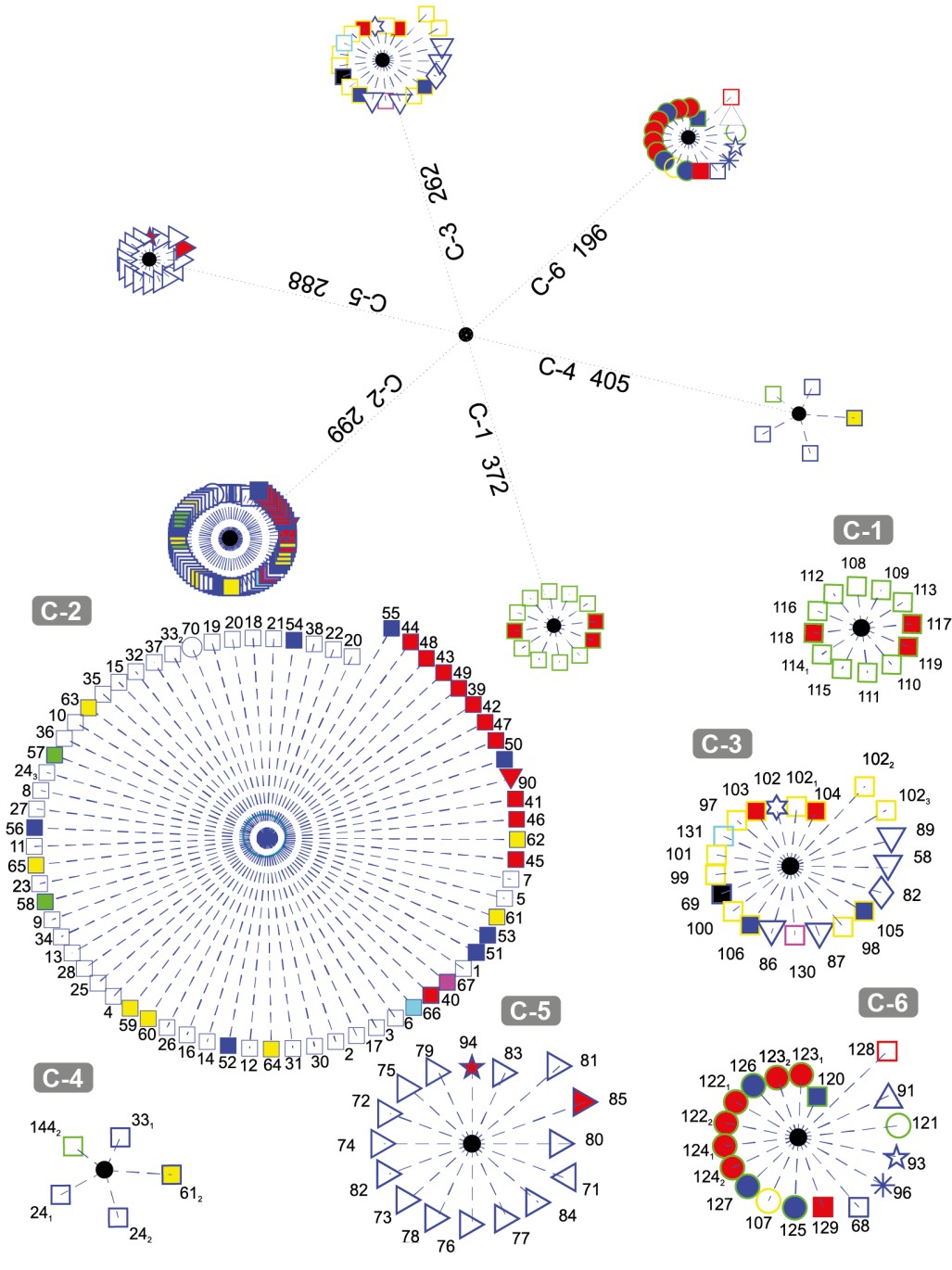

**Figure 3** DNA clustering for marker COXI with $k = 6$.

are smaller in comparison with the other organisms and their corresponding centroids, which indicate that arthropods in the selected dataset are all very far away from every other organism, something that is consistent with the *Hebert, Ratnasingham & de Waard (2003)* findings on COXI divergence analysis.

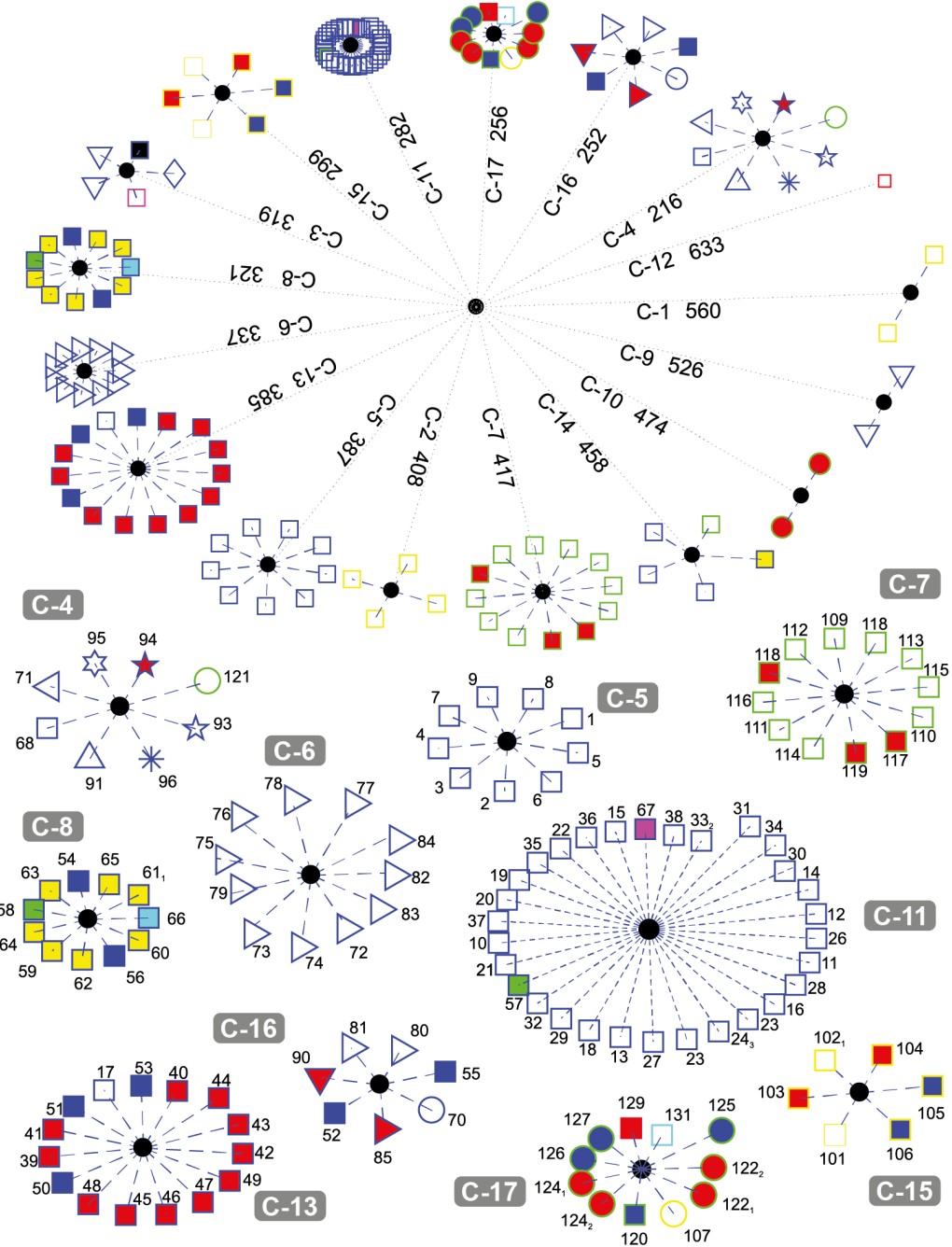

**Figure 4  DNA clustering for marker COXI with $k = 17$.**

Figure 4 depicts the results obtained when the dataset was grouped into 17 clusters.

Note that cluster C-6 is a refined version of C-5 in $k = 6$, since one of the Cnidarians the Fresh-water Polyp *Hydra vulgaris* (red-fill star), the Branchipod Water Flea *Daphnia pulex* (red-fill right-facing triangle), and the Hemichordate Acorn worm *Saccoglossus kowalevskii*

(left-facing triangle) have moved to other clusters, leaving only insects in this group. The two Cnidarians and the Hemichordate are now together in C-4, while the Branchipod is in C-16

Note that for this value of $k$, fungi grouped into three well-defined clusters (C-1, C-2, and C-15), with the exception of the Basidiomycota Corn smut *Ustilago maydis* (yellow-edge circle) which keeps its grouping with other organisms. It is interesting that the two members of C-1 are two of the copies of *Yarrowia lipolytica* (yellow-edge square), while the other fungi in C-15 are of heterogeneous classes.

The Tracheophyta plants cluster C-7 remains with the same organisms of classes Magnoliopsida (green-edge squares) and Liliopsida (green-edge red-fill square), while the group of the Chlorophyta plants separated the two copies of the Picoplanktom *Bathycoccus prasinos* (green-edge red-fill circle) that end up together in C-10, and the Picoplankton *Chlamydomonas reinhardtii* (green-edge circle) which is grouped in C-4 with other organisms. The second copy of the Castor oil tree remained with the same organisms in C-14, which is exactly the same as C-4 in the $k = 6$.

The fact that the two copies of the Picoplankton *B. prasinos* (C-10) are both clustered together apart from the other plants is because they are either very recent orthologue duplications or have not been verified accurately, as they have both the same sequence entry in NCBI database (SequenceID: NC_023273.1) reported at different loci in its mitochondrial genome (GeneID: 18158061 and GeneID: 18158101).

Chordates are separated into four clusters (C-5, C-8, C-11, and C-13) with all Hominids grouped together in C-5. C-8 is formed by two reptiles, one anfibious, and the fish, both Actinopterygii (blue-edge yellow-fill square) and Sacropterygii (blue-edge cyan-fill square), C-13 is formed with all the birds (blue-edge red-fill square), some reptiles (blue-edge blue-fill square), and the Naked Mole-rat *Heterocephalus glaber*, and C-11 with the rest of the Chordates in a very compact group. Note that the result of the birds grouped with the reptiles may be explained by the evolutive theory that claim that the birds are descendant of ancient saurid reptiles. It is also interesting that reptiles tend to group with other organisms and not necessarily between them. This could be the result of high diversity of COXI among reptiles, as reported by *Vasconcelos et al. (2016)*.

Cluster C-9 contains two Secernentea patogens of the Onchocercidae order, Lymphatic Filaria *Brugia malay* and Eye Filaria *Loa loa*, both parasites of humans and other animals and have a clear evolutive difference defined by the enviroment in which they live in, compared with the other two Secernentea of the free-living Rhabditida order *Caenorhabditis elegans* and *Caenorhabditis briggsae*. Our results agree with those of *Prosser et al. (2013)*, where successfull COXI operational taxonomic units were developed to differenciate between parasitic and free-living taxa.

Note that one of the organisms with no assigned kingdom, the Cyanoalgae thermoacidophillic *Galdieria sulphuraria* generated its own cluster C-12.

It is interesting that some clusters are more compact than others (e.g., C-11 and C-6, vs. C-13 and C-7). The compactness of a cluster indicates the degree of relationship of the organisms belonging to it with respect to a common reference (i.e., how similar they are between them).

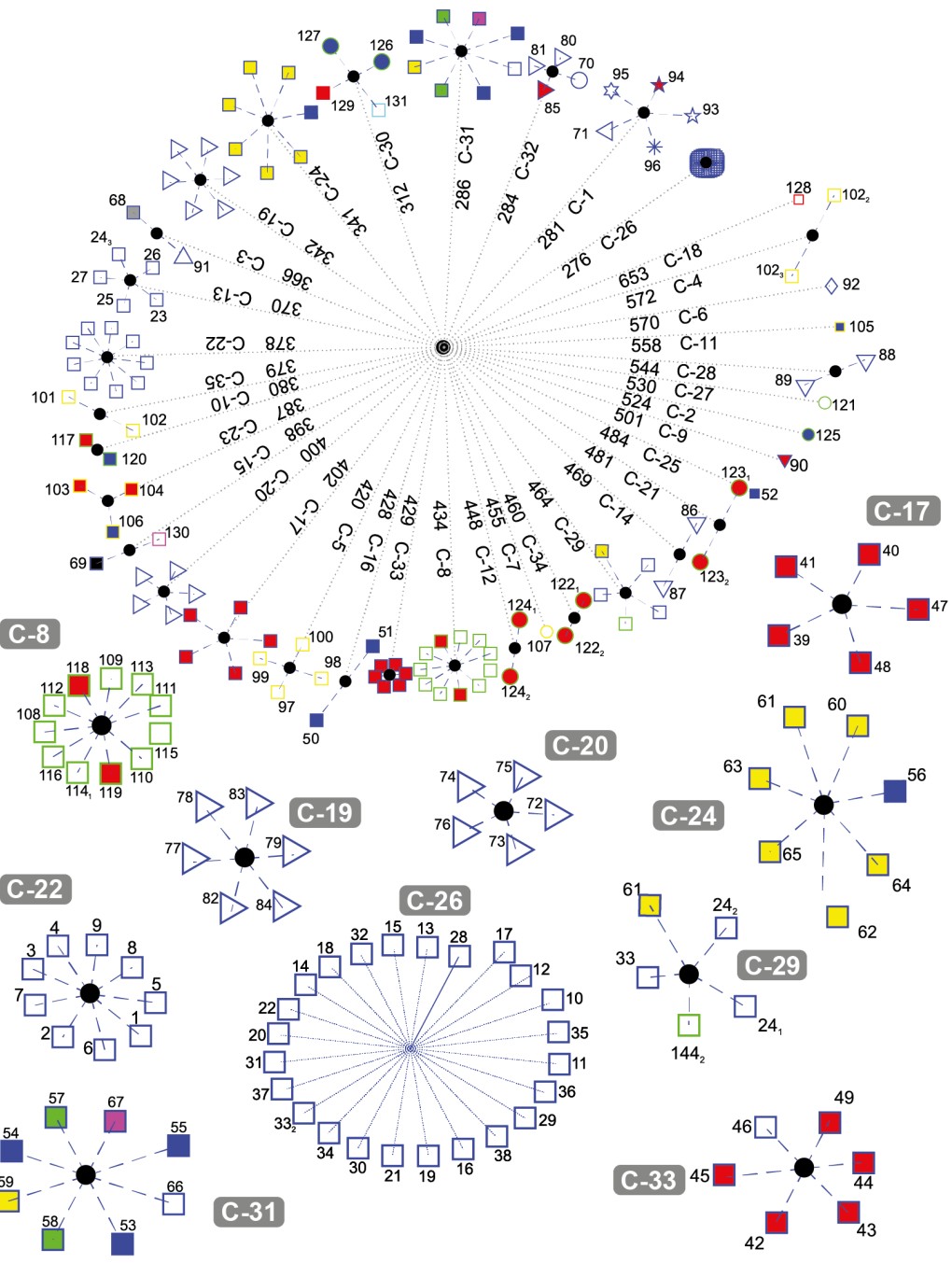

**Figure 5  DNA clustering for marker COXI with $k = 35$.**

Figure 5 depicts the results obtained when the dataset was grouped into 35 clusters. Note that with some exceptions, most of the data in $k = 35$ is more clearly clustered together by their respective class and some were downright to their order or even lower phylogenies. For the plants, new clusters were generated: C-12 with the two copies of Pikoplanctom *M. commoda*, C-34 with the two copies of Picoplancton *Ostreococcus tauri*

in the other, C-10 with the Japanese rice *Oryza sativa japonica* and the Moss *Physcomitrella patens*, C-27 with the Picoplankton *C. reinhardtii* (which previously was the only plant clustered in C-4 for $k = 17$). Originally in $k = 6$, Picoplanktons were grouped together with other organisms, but isolated from the other plants. At this level of cluster decompositio, we can observe that Picoplanktons are all separated, probably because they are unicellular organisms and will present large variation in the COXI marker (*Lin et al., 2009*).

C-17 and C-33 are well defined clusters of birds. C-20 is a group of flies from the Diptera class, the remaining non-fly Diptera, Lepidoptera and Hemiptera are grouped in C-3, while C-32 includes the Red Flour Beetle *Tribolium castaneum* and the Red Fire Ant *Solenopsis invicta*.

C-13 contains five out of the seven Artiodactylians, C-22 corresponds to the hominidies which did not change since $k = 6$ and the first and closest compact cluster C-26 corresponds to all of the remaining mammals.

A very interesting feature is that C-29 is the same multi-class cluster that appeared in $k = 17$ and $k = 6$ conformed by the Bat *M. davidii*, both Yak *B. grunniens* copies, the second Castor oil tree *R. communis* copy, and the Spotted green pufferfish *T. nigroviridis*. When we explored the characteristics of those gene sequences, we found that all of them are significantly below the average gene size $1,545.8 \pm 124.5$ bp. The NCBI database reported that all of them are not mitochondrial genes, but the product of nuclear genomic sequencing where scaffold primary assembly showed those fractions with alignment homology reported to COXI, but not proven genetic activity. We also found that both the second and third copies of *Y. lipolytica* in C-4 are significantly above the average COX1 gene size. These last two genes correspond to coxI-i5 and coxI-i7 that contain unusually large exons 5 and 7 respecively (NCBI GeneID: 802596; Sequence entry: NC_002659.1), which gives them the extra sequence length in the KEGG database.

The two Alligators *A. sinensis* and *A. mississippiensis* (blue-edge blue-fill squares) generated their own cluster in C-16. The fellow Reptiles, the Green Sea Turtle *Chelonia mydas*, the Burmese Python *Python bivittatus* and the Lizzard *Anolis carolinensis*, clustered together with both frogs *Xenopus laevis* and *Silurana tropicalis*, the Zebrafish *Danio rerio* and the cartilaginous fish Australian ghostshark *Callorhinchus milii* (blue-edge magenta-fill square) in C-30, leaving C-19 as a better defined cluster with most ray-finned fish and only the *Gekko japonicus* barging in the group. The Chinese Softshell Turtle *Pelodiscus sinensis* created its own cluster in C-25.

Clusters C-18, C-6, C-11, C-27, C-2, C-9, C-25 and C-7 are one-organism clusters. That may be explained because these organisms are the most external with respect to their classes or phyla. For instance, in C-18 we find the Cyanoalgae thermoacidophillic *G. sulphuraria*, while in C-6 we find the most outside group of the Metazoans that correspond to Porifera phyla, the Sponge *Amphimedon queenslandica*. Moreover, $k = 35$ cluster distance spans from 276 to 653; just before the second half of the average distance, at 448 lies all of the lone clusters and most of the two-sequence cluster with the sole exception of the unpaired COX1 gene sequence size C-29.

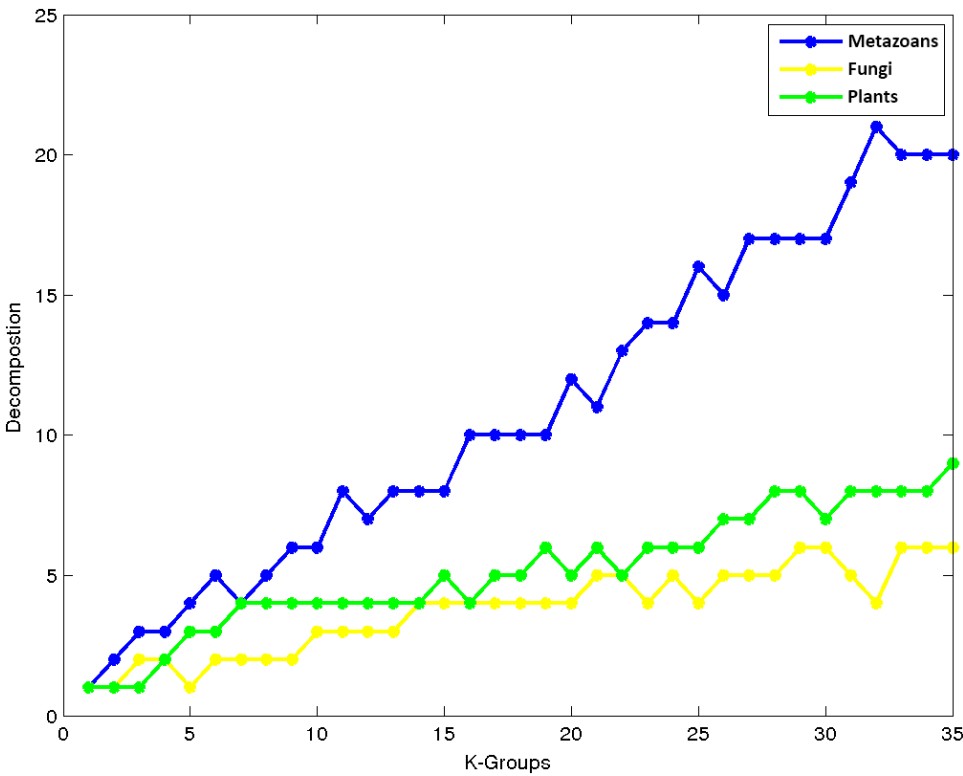

**Figure 6** **K-means decomposition analysis.**

An interesting property observed in our experiments is that as we increase the number of groups, the data corresponding to the kingdoms are separated at different rates. Figure 6 depicts how the kingdoms are decomposed into a number of groups with respect to the number of clusters. Note that Metazoans separate faster as compared to Fungi and Plants. This may be explained by the large number of organisms belonging to this kingdom which have a greater chance to group together due to their high class similarities. Note that the second largest kingdom of Plants decompose faster than Fungi, which is the third largest group.

To determine the validity of the results, we computed centroids for true kingdoms and we compare these centroids to those discovered with our method. Figure 7 depicts the mean square distances between each cluster centroid and the sequences assigned to that cluster by the proposed method using $K = 6$, and the mean square distances between a cluster centroid generated with the sequences corresponding to each of the six kingdoms were compared. Note that the centroid of sequences belonging to the plant kingdom has a large similarity with respect to C-1 which contains most of the plants. Similarly, the Metazonans kingdom have a large similarity with C-2 which is conformed by the majority of the Chordates. The Fungi kingdom depicts a large similarity with respect to C-3 which contains most of the fungi. Moreover, note that the other kingdoms depict a larger distance with respect to the all the clusters.
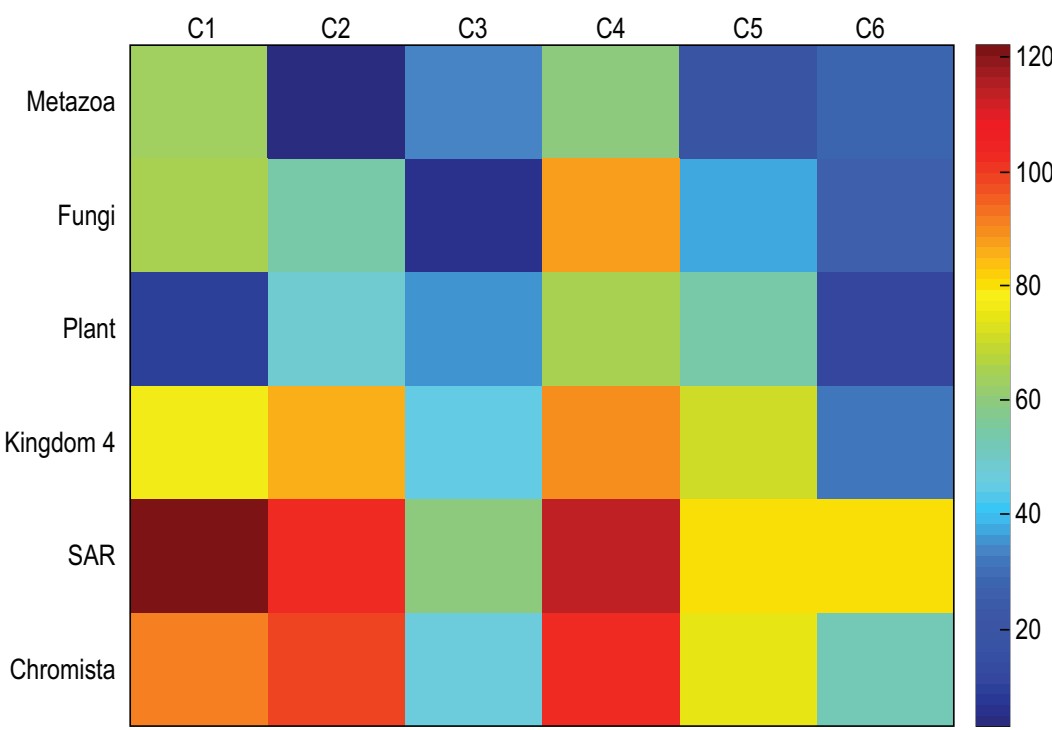

**Figure 7** Mean square distances between each cluster centroid and the sequences assigned to that cluster by the proposed method using $k = 6$, and the mean square distances between a cluster centroid generated with the sequences corresponding to each of the six kingdoms.

## Comparison with other cluster methods

We evaluated the performance of the MATLAB implementation of proposed algorithm "Signal Tool for the Analysis of the Relationship between Sequences" (i.e., STARS) in terms of computational time with respect to ClustalW and UCLUST. While ClustalW is not strictly a clustering method, we used it for comparison because it is one of the most commonly used tools to evaluate the similarity of multiple sequences. We employed a CPU Intel XEON E5-1650 at 3.50 GHz with 16 GB RAM.[2] Table 1 list the processing time in seconds for the three methods for sets of 8, 17, 35, 70, and 141 sequences of COXI. The time required to transform the 141 sequences from strings of characters to their corresponding PSDs was 0.921 s and it is not considered in Table 1 since this is performed only one time. Note that the time required by STARS is significantly smaller with respect to ClustalW. UCLUST is time-constant at 1 s for every experiment, however, note that the number of clusters generated by this method was practically the same number of sequences (i.e., the method assigns a cluster to each sequence). This is because UCLUST requires a sequences identity range of at least 40% for amino acids and 65% for nucleotides (*Edgar, 2010*).

Table 2 list the processing times of five datasets of different number of sequences with different length where UCLUST generated a number of clusters different from one cluster for each sequence. Dataset A consisted of 31 sequences of Mammals with average length of 16,695 nucleotides labeled into to seven groups; Dataset B consisted of 38 sequences

[2]When analyzing the processing times of the compared methods, it is important to consider that the STARS was implemented in MATLAB without any parallelization, in comparison with the highly optimized implementations of ClustalW and parallelized UCLUST.

**Table 1** Performance comparison of STARS with respect to ClustalW and UCLUST on sets of different sizes of COXI sequences.

| Number of sequences | STARS $K = 6$ (s) | STARS $K = 17$ (s) | STARS $K = 35$ (s) | ClustalW (s) | UCLUST (s) / No. of resulting clusters |
|---|---|---|---|---|---|
| 8 | 0.034 | – | – | 2.8 | 1 / 8 |
| 17 | 0.038 | 0.071 | – | 9.85 | 1 / 17 |
| 35 | 0.071 | 0.098 | 0.167 | 32.68 | 1 / 35 |
| 70 | 0.160 | 0.188 | 0.279 | 130.91 | 1 / 70 |
| 141 | 0.383 | 0.655 | 0.770 | 485.02 | 1 / 138 |

**Table 2** Performance comparison of STARS with respect to UCLUST on sets of different sequences.

| Dataset | Number of sequences | Average sequence length | Number of clusters | Sequence to PSD transform (s) | STARS (s) | UCLUST (s) |
|---|---|---|---|---|---|---|
| A | 31 | 16,695 | 6 | 1.92 | 0.70 | 1 |
| B | 38 | 1,407 | 4 | 0.27 | 0.03 | 1 |
| C | 116 | 7,154 | 17 | 3.05 | 1.69 | 1 |
| D | 34 | 27,567 | 12 | 3.38 | 1.23 | 12 |
| E | 30 | 3,361,393 | 8 | 392.98 | 281.70 | – |

of Influenza A viruses with average length of 1,407 nucleotides labeled into to six groups; Dataset C consisted of 116 sequences of Human Rhinovirus with average length of 7,154 nucleotides labeled into four groups; Dataset D consisted of 34 Coronavirus sequences with average length of 27,567 nucleotides labeled into six groups; and Dataset E consisted of 30 sequences of Bacteria with average length of 3,361,393 labeled into eight groups. Note that the computational time required for performing the clustering of the sequences' PSD data is smaller when compared with UCLUST for the same number of clusters for datasets A, B, and D. In the case of dataset E, we could not achieve a result using UCLUST (i.e., the program throws a fatal error) indicating that the data was too big.

## DISCUSSION

Numerous reports have discussed the use of different molecular markers to determine the appropriate phylogenetic divergence at many levels of the tree of life. For our experiments, we considered molecular markers previously employed in phylogenetic analysis for the evaluation of the differentiation capability of DNA sequences (*Hoang et al., 2015*) (e.g., COXI, mtDNA, influenza A virus, human rhinovirus, coronavirus and bacterial genomes). In this work, we focused on the Mitochondrial Cytochrome C Oxidase Subunit I (COXI) coding gene as a marker to evaluate our approach for group clustering of relevant similar sequences. The COXI gene has been proposed as one of the most relevant marker genes for molecular taxonomy (*Patwardhan, Ray & Roy, 2014*). While no single gene is even close to establishing the systematic classification of organisms, the COXI gene is one of the most closely related to the consensus evolutionary divergence. Therefore, it is important to wrap our results not as how the Catalogue of Life (*Roskov et al., 2017*) classification should become, but as how accurate our marker could be. The selection (COXI) was based on

three criteria: (i) the marker must code for proteins since it has been already proven that these type of markers have steadier mutation rates, (ii) the marker should have already been employed in a wide range of the tree of life, at least for eukaryotes, and should be able to discriminate for the intended groups, and (iii) the marker should have a homogeneous length and have a minimum number of reported copies in the selected database, since both duplication events and large indels may bias new cluster formation. To rule out any bias in the clustering of organisms with respect to their downloaded sequences, we incorporated all of their stored copies.

A possible expected result was that the clusters generated for each selected value of $k$ would correspond to the organization depicted in Fig. 2. However, since the K-means method promote the generation of centroids in highly populated regions of the feature space, it is more likely to obtain clusters of organisms that are highly related among them, instead of organisms related by possible common ancestors or groups with a small number of less homogeneous organisms (e.g., primates formed a cluster early in small $k$ values and kept together at larger numbers of $k$).

The COXI gene is one of the most accepted general markers to establish divergence (*Patwardhan, Ray & Roy, 2014*). It spans from Phylum to Class, and when using introns in selected species, it has been shown to properly classify Genera and Species (*Zardoya & Meyer, 1996*). Our results were remarkably good in clustering up to the Family level by using only the coding region and without the need to pretreat or manually curate the sequences.

*Hebert, Ratnasingham & De Waard (2003)* established a divergence rate from 0.01% to 64% with a median of 8% across a number of species on 11 Metazoan phyla. In that study, Arthropod (with the exception of Lepidoptera and Diptera class) and Plathelminth phyla displayed the greatest divergences, while Chordata showed the second lowest divergence. Our results showed high cohesiveness, particularly for Chordates, where they quickly established stable, compact clusters, predominantly with their own classes. Since the downloaded data for each phylum or class was not balanced, we had the opportunity to evaluate how the sequences are clustered in a real-life condition. For example, when sampling whole ecosystems (i.e., microbiomes), bacterial populations will not be balanced across their species, but will show predominant phylogenetic diversity toward certain groups. Our results show that our presented method is very sensitive to both, the relative abundance of tight clusters, and the $K$-number. Far from being a disadvantage, we found that changing the number of clusters in an experiment may provide new insights about the relationship between the various sequences.

Sequence mutations of COXI coding regions have not been shown to distribute bias towards any segment or region, something like what happens to other markers such as the ribosomal 16S gene, where changes on highly conserved regions are very few and slow, while changes on hypervariable regions show rapid changes that can determine divergence along several phylogenetic groups, according to which hypervariable region is being evaluated. If this would be the case, $K$-Means clustering may be adapted to steps of low mutation rates before high mutation rate regions. COXI gene mutations spanning all of the sequence may increase the amount of spurious clustering due to converging hotspots.

The presence of a spurious cluster that is gathered together by their size, is an indication of the need to filter out sequences with large indels. Despite such mishaps, the proposed method is capable of performing an analysis of relationships between multiple DNA sequences with minimum handling and without the need of sequence alignment, which results in less human and computational time compared to traditional methods. We tested this method with a number of markers (i.e., mammal mtDNA, influenza A virus, human rhinovirus, coronavirus, and bacterial genomes) previously employed for Fourier DNA spectra phylogenetic analysis (*Hoang et al., 2015*), the results are shown in Supplemental Information 1 of this article. Briefly, most sequences evaluated under our method cluster properly and consistently with previous reports (*Hoang et al., 2015*). Also consistent with COXI results, the most evident aspect is the tendency to prioritize division of heavily populated groups.

The proposed method may be used to evaluate the capability of a marker or gene to differentiate between organisms at different levels, to identify subgroups within a set of organisms, and perform classification of organisms with respect to known sequences or classification of sections of a DNA sequence. Furthermore, this method can also be used to perform similar analysis with amino acid sequences.

We have demonstrated that it is possible to group DNA sequences based on their frequency components. It is the subject of future work to identify whether distinct frequency bands amount to greater weight in the clustering of sequences.

The proposed method has been coded and executed in MATLAB. The source code and the datasets employed for the results presented in this paper are available at Github

## CONCLUSION

We have presented a method for performing cluster analysis of DNA sequences that is based on the use of GSP methods and the K-means algorithm. We also proposed a visualization method that allows us to easily inspect and analyze the results and possible nontrivial relationships. Our results indicate the feasibility of employing the proposed method to find and easily visualize interesting features of sets of DNA data.

## ACKNOWLEDGEMENTS

The authors thank CONACyT and PRODEP for the provided support. Any opinions, findings, conclusions or recommendations expressed in this material are the sole responsibility of the authors and may not reflect the views of the sponsors.

### Funding
The authors received no funding for this work.

### Competing Interests
The authors declare there are no competing interests.

## Author Contributions

- Gerardo Mendizabal-Ruiz conceived and designed the experiments, performed the experiments, analyzed the data, wrote the paper, prepared figures and/or tables, reviewed drafts of the paper.
- Israel Román-Godínez and Ricardo A. Salido-Ruiz conceived and designed the experiments, performed the experiments, analyzed the data, contributed reagents/materials/analysis tools, wrote the paper, reviewed drafts of the paper.
- Sulema Torres-Ramos conceived and designed the experiments, performed the experiments, contributed reagents/materials/analysis tools, wrote the paper, reviewed drafts of the paper.
- Hugo Vélez-Pérez conceived and designed the experiments, contributed reagents/materials/analysis tools, wrote the paper, prepared figures and/or tables, reviewed drafts of the paper.
- J. Alejandro Morales conceived and designed the experiments, analyzed the data, wrote the paper, prepared figures and/or tables, reviewed drafts of the paper.

## Data Availability

Github: https://github.com/starsudg/STARS.git.

## Supplemental Information

Supplemental information for this article can be found online at http://dx.doi.org/10.7717/peerj.4264#supplemental-information.

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
