# Peer review of "Genomic signal processing for DNA sequence clustering"

_PeerJ, doi:10.7717/peerj.4264_

## Round 0.1 · original submission · Major Revisions

Both reviewers (and I) found your manuscript to be clearly written and to report a useful advance in sequence clustering. The reviewers make several suggestions that would improve the manuscript, though.

Following are the reviewers' suggestions that I would like you to address in a revised manuscript:

1. Both reviewers request that you deposit your source code in an open source repository (such as GitHub), so that it is accessible to the community.

2. To support the claim that your method is faster than other methods, both reviewers ask that you benchmark your method against ClustalW and UCLUST and provide performance data.

3. While the reviewers find that your clustering results for COX1 are interesting, they both suggest that you apply your approach to additional genes to support the general validity of your approach. This may be appropriate for submission as supplementary material.

4. Reviewer 2 suggests that you apply your method to a gene for which phylogeny is well established, as a kind of control and test of the validity of your approach. In the same vein, Reviewer 1 suggests that you compute centroids for true phyla, etc. and compare these centroids to these you discovered with your clustering method, again as a test of the validity of your results.

5. Both reviewers provide many suggestions for improving the presentation of the paper (typos, style, figure labelling).

Reviewer 1 ·

Basic reporting

Overall, the paper is well-written and easy to understand. There are a few minor spelling and grammatical errors present in the manuscript that the authors should correct. Here are some examples: 
1 Line 13 should read "...popular method that has been used to..." 
2 The sentence on lines 33-35 should read "However, the K-means algorithm relies on the frequent computation of similarity metrics.." 
3 The reference on line 159 is incorrectly formatted. 
4 The algorithm's name, STARS, is misspelled as SATRS on page 5 of the installation guide and page 1 of the user manual. 
5 The title is shown as "Genomic signal processing for DNA sequence clustering" and "Genetic signal processing for DNA sequence clustering." I suggest using the former, as it reflects the wording in the main body of the text.
6 The word "Matlab" on line 322 should be stylized as MATLAB
I also would like to make the following minor suggestions (starting from most important to least important):
1 The figures have very small font, making them very hard to read. I suggest that the authors increase the font across all figures, especially for Figures 3-6, which have a lot of unused white space.
2 According to the user manual in the link provided by the authors, the name of the algorithm is STARS. However, this name does not appear anywhere in the manuscript. The authors should include the algorithm's name at least once in the manuscript.
3 The first two lines of the abstract (11-12) and the introduction (23-24) are exactly the same word-for-word. For some readers, this may come off as repetitive and distracting. I suggest that the authors change the wording of either the openings of the abstract or introduction.
4 I suggest removing the word "novel" in the sentence on lines 58-60.
5 Lines 303-309 are a little confusing to me. I am not exactly sure what the authors mean by phylogenetic trees requiring an external class. Either the authors should clarify this statement or remove the paragraph altogether.
I have one major comment for this section:
The papers "A new method to cluster DNA sequences using Fourier power spectrum" (Hoang et al., 2015) and "A novel clustering method via nucleotide-based Fourier power spectrum analysis" (Zhao et al., 2011) also cluster DNA sequences by first computing power spectra and then converting spectra to constant length vectors. I suggest that the authors cite these works and describe what sets their method apart from these previous methods. If possible, the authors should include a comparison in terms of running times and accuracy against these methods.

Experimental design

The authors describe their method with sufficient detail for others to replicate. Briefly, they convert DNA sequences to power spectra, and then group DNA sequences together using the K-means clustering algorithm applied on the frequency components of the power spectra. In order to make the power spectra of varying length DNA sequences comparable, they apply zero padding, as suggested by Borrayo et al., 2014. They applied their method to a set of 141 COXI sequences from several different organisms. I have the following major comments for the author that I believe will help improve the manuscript:

1 The authors claim in lines 50-57 that sequence alignment algorithms like ClustalW and UCLUST are limited in the number of sequences they can cluster due to their heavy demand of computational resources. However, the authors do not provide a comparison of their algorithm with any of these algorithms, nor do they mention the running time of their algorithm. I would like to see a comparison against other algorithms like ClustalW and UCLUST that includes running time, memory usage, and/or clustering quality to justify their claims.
2 As stated in lines 322-324, the authors have made their program as a standalone package. I commend the authors for making their algorithm readily available for users. However, I was unable to actually test their algorithm since it could only run on Windows. Most bioinformatics researchers use either Linux or Mac OS X, and hence replicating the authors' results would be difficult for such readers. The authors should provide the code open source (preferably on Github) so that users can run the code on their own copies of MATLAB or Octave.
3 As depicted in Figure 2, hierarchical organization of the selected organisms plays a very important role in the manuscript's analysis. The hierarchical clustering algorithm is an alternative to the k-means clustering algorithm, and seems like a more appropriate fit for phylogenetic studies. I suggest that the authors repeat their experiments using hierarchical clustering and report the results in the manuscript.
4 The authors should explicitly state how many frequency components each COXI DNA sequence is converted to (i.e. the value of "n" in line 103).

Validity of the findings

The authors applied their method on COXI sequences from various species and find that the resulting clusters reflect known biology. They also provide a thorough discussion of their results. 

1 I would like to see the authors repeat their experiments for sequence sets other than COXI. The Hoang et al. 2015 paper, for example, applies their clustering algorithm on 5 different datasets. Perhaps the authors can include the analysis on other sequence sets as a supplementary. 
2 Figures 3-5 show clusterings for the COXI set for various values of k. The values of k are chosen to reflect kingdoms, phyla, and classes. However, the clusters do not exactly correspond to these taxonomic ranks. The authors should compute centroids corresponding to the "true" kingdoms, phyla, and classes and compare the "true" clusters with the discovered clusters. A comparison of the inertia, or the within-cluster sum of squares criterion, between the "true" and discovered clusters should suffice.

Reviewer 2 ·

Basic reporting

In this manuscript, the authors present a method for clustering
sequences in which the K-means algorithm is applied to a numerical
representation (the Voss representation) of DNA sequences to which
spectral analysis (a discrete Fourier transform) is applied.

The authors also present a visualization method to report the
resulting clustering.

Data sharing
* * *
- Link to the method is mentioned (lines 322-324) but not provided. Thus, I could not test the method.

- Datasets not provided

- The authors mention that they will provide "... executable and the dataset" (line 323). The executable
is not enough. They need to provide the source code as well. Without the source code, the paper should
not be published.



Professional English
* * *
The article reads well.

Only comment: Avoid repetition of whole sentences. The first two
sentences at the beginning of the Abstract are identical word-by-word to
those at the beginning of the Introduction.


Self contained
* * *
In order to fully understand the methods, it would be relevant to have a full definition
of the

"power spectral density (PSD) {\hat S^\alpha}}" (lines 86-87),

as well as the

"frequency power spectra d(Sa,Sb)" (lines 89-90)

Experimental design

The authors apply their method to one gene: Cytochrome c oxidase I
(COXI), using different numbers of clusters. It is unclear why this
particular gene was selected, and why only one was analyzed.

The authors make a point of this method being fast. I would like to
see a experiment in which they show how much faster they are, compared
to, for instance, the other two methods they mention as slow: CD-HIT
and UCLUST

Validity of the findings

The description of what is found in the different clusters for COXI
is not very informative, as we do not know what the ground truth is.

Any clustering algorithm, is going to produce clusters. Assessing the
validity of the method presented in this paper, would require some
proof of principle. I would like to see the analysis of a gene for
which the phylogeny is well established (as different methods agree),
and how that compared with the method presented here.

Additional comments

Here I report some small changes that could improve the manuscript

- Abstract: GPS abbreviation used prior to definition

- All Figure need a more descriptive legend, and explanation of terms.

- Figure2: name of species are not readable

- Figures 3, 4, and 5: labels are not readable.

---

## Round 0.2 · Minor Revisions

Thank you for providing new analyses of five additional datasets in a supplement, as requested by the reviewers. However, these analyses (and the existence of the Supplement) are not mentioned in the main text of the manuscript. Please add a paragraph that describes the supplementary analyses and briefly summarizes the results.

Reviewer 1 makes a few suggestions for improving grammar and spelling. I also noted a few places in the manuscript that need to be corrected:

L 71-72:
Cluster analysis of DNA signals trough the use of GSP methods have been previously proposed (Zhao et al. (2011); Hoang et al. (2015). However, these methods are based are based on the computation 


(change to "through"; "are based" is repeated)

L 80:
allows up to easily inspect and analyze the results 


(change "up" to "us")

L 243:
the Green Sea Turtle Chelonia mydas, the Burmese Phyton 


(change to "Python")

L 264:
these centroids to these you discovered with our method. 


(change to "to those discovered with our method")

L 319-320:
(e.g., primates formed a cluster early in small k values and kept together at larger numbers of k. 


(missing parenthesis at end of sentence)

Reviewer 1 ·

Basic reporting

Grammar and spelling is still a minor issue in this paper. Here are a few examples from the added text:
A) "To determine the validity of the results, we computed centroids for true kingdoms and we compare these centroids to these you discovered with our method.” (263-264)
B) "UCLUST have a constant time, of 1 second for every experiment…” (282-282)
3) "However, since the K-means method promote the generation of centroids in highly populated regions of the feature space it results are more likely to obtain clusters of organisms that are highly related among them instead of organisms related by possible common ancestors or groups with a small number of less homogeneous organisms" (317-320)

Experimental design

no comment

Validity of the findings

no comment

Reviewer 2 ·

Basic reporting

This revision has taken into account most of the questions raised by the reviewers.
I do not have any further comments.

Experimental design

no comment

Validity of the findings

no comments

Additional comments

no additional comments

---

## Round 0.3 · accepted · Accept

Thank you for submitting this very interesting study to PeerJ.